# Piezomagnetism and magnetoelastic memory in uranium dioxide

M. Jaime[1], A. Saul[2], M. Salamon[1], V.S. Zapf[1], N. Harrison[1], T. Durakiewicz [1,3], J.C. Lashley[4], D.A. Andersson[5], C.R. Stanek[5], J.L. Smith[6] & K. Gofryk[7]

The thermal and magnetic properties of uranium dioxide, a prime nuclear fuel and thoroughly studied actinide material, remain a long standing puzzle, a result of strong coupling between magnetism and lattice vibrations. The magnetic state of this cubic material is characterized by a 3-**k** non-collinear antiferromagnetic structure and multidomain Jahn-Teller distortions, likely related to its anisotropic thermal properties. Here we show that single crystals of uranium dioxide subjected to strong magnetic fields along threefold axes in the magnetic state exhibit the abrupt appearance of positive linear magnetostriction, leading to a trigonal distortion. Upon reversal of the field the linear term also reverses sign, a hallmark of piezomagnetism. A switching phenomenon occurs at $\pm 18$ T, which persists during subsequent field reversals, demonstrating a robust magneto-elastic memory that makes uranium dioxide the hardest piezomagnet known. A model including a strong magnetic anisotropy, elastic, Zeeman, Heisenberg exchange, and magnetoelastic contributions to the total energy is proposed.

[1] MPA-CMMS, Los Alamos National Laboratory, Los Alamos, NM 87545, USA. [2] Aix-Marseille University, CINaM-CNRS UMR 7325 Campus de Luminy, Marseille cedex 9 13288, France. [3] Institute of Physics, Maria Curie-Sklodowska University, PL-20-031 Lublin, Poland. [4] MPA-11, Institute for Materials Science, Los Alamos National Laboratory, Los Alamos, NM 87545, USA. [5] MST-8, Los Alamos National Laboratory, Los Alamos, NM 87545, USA. [6] Sigma Division, Los Alamos National Laboratory, Los Alamos, NM 87545, USA. [7] Idaho National Laboratory, Idaho Falls, ID 83415, USA. Correspondence and requests for materials should be addressed to M.J. (email: mjaime@lanl.gov) or to A.S. (email: saul@cinam.univ-mrs.fr) or to K.G. (email: krzysztof.gofryk@inl.gov)

E xperimental studies carried out on $UO_2$ under the cloak of the Manhattan Project showed the first hints of what later came to be accepted as antiferromagnetism (AFM) at $T_N = 30.8$ K[1, 2]. Extensive neutron-scattering measurements revealed non-collinear spin ordering with a 3-**k** structure below $T_N$[3-5]. This is accompanied by a static Jahn-Teller distortion of the oxygen cage, and strong magnetoelastic interactions[6-9] that emerge from a face-centered cubic (fcc) structure (see Fig. 1a, b). It was argued that a large third-order invariant in the free energy expansion that couples magnetic dipoles and electric quadrupoles[6, 10, 11] results in the first-order nature of this magnetoelastic transition. A dynamic Jahn-Teller model was also proposed[12] to explain the persistence of strong magnetoelastic coupling well above $T_N$. Owing to the symmetry of the non-collinear 3-**k** antiferromagnetic (AFM) order in $UO_2$, the existence of piezomagnetism (PZM) is possible[13] but has never been observed. PZM, first predicted by Dzyaloshinsky[14, 15], is characterized by a

linear coupling between the system's mechanical strain and magnetic polarization. In PZM crystals, a magnetic moment can be induced by application of a physical stress, and it has captured attention in recent years as a mechanism that could be used in combination with multiferroics and piezoelectrics at the nanoscale to achieve control of magnetism by electric fields[16]. Among the 122 space groups that describe magnetic order, only a subset of 66 can present PZM. It is precluded in the remaining 56 because they contain time reversal as a symmetry element (32 groups), its product with inversion (21 groups), or have spatial-only symmetries incompatible with the axial character of the PZM tensor (3 groups).

Here we have uncovered a magnetostriction (MS) linear in field, the converse of PZM[17], that confirms the non-collinear 3-**k** nature of the magnetically ordered state in $UO_2$. High coercive fields of 18 T were found when the direction of applied magnetic field is reversed, making it the hardest piezomagnet

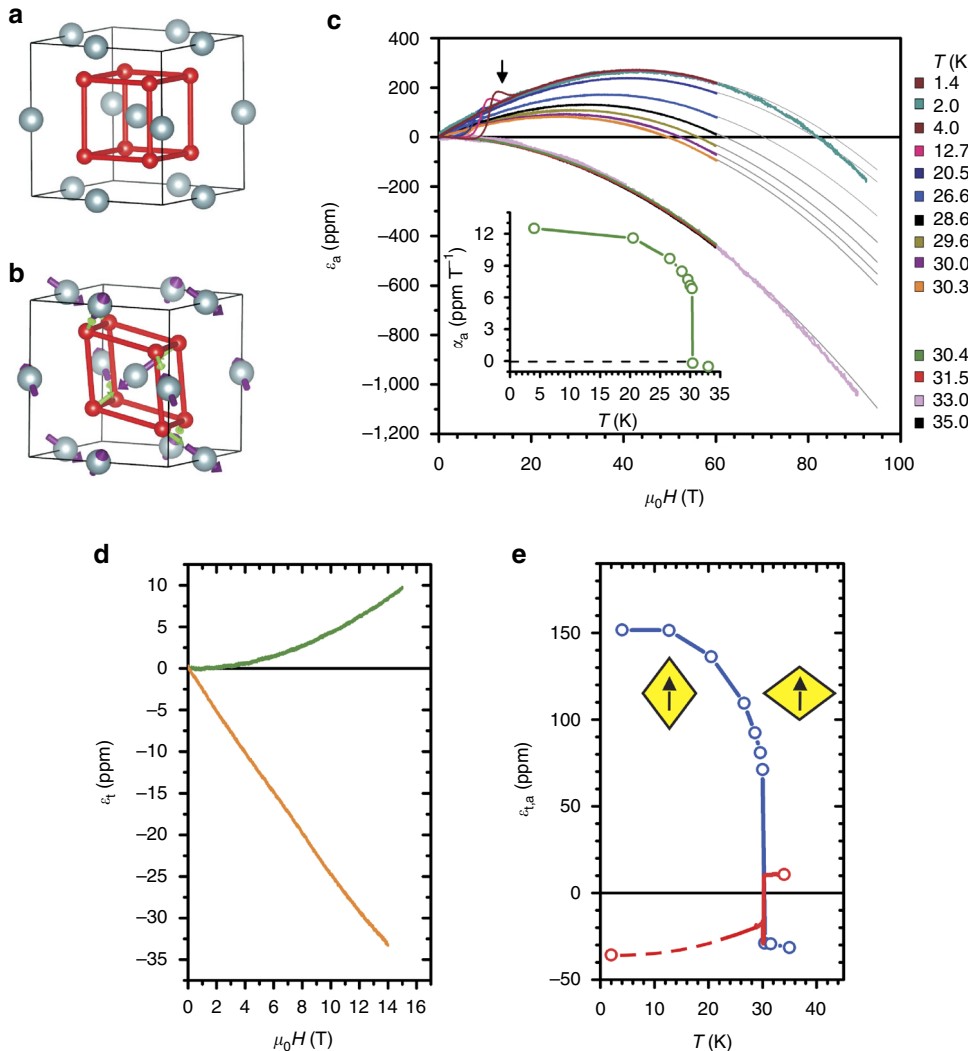

**Fig. 1** Axial and transverse magnetostriction in $UO_2$. **a** fcc unit cell of $UO_2$. **b** The low-temperature antiferromagnetic state displaying the transverse 3-**k** $T_A$ magnetic order (*violet arrows*) and oxygen displacements (*green arrows*, not to scale) along the $\langle 111 \rangle$ directions. **c** Isothermal axial strain $\varepsilon_a$ vs. **H** parallel to [111] measured in pulsed magnetic fields to 92.5 T at different temperatures listed on the *right-hand side*. *Grey lines* are fits to the expression $\alpha_a H + \beta_a H^2$. The jump between the curves for 30.3 and 30.4 K is a consequence of the first-order phase transition. Linear magnetoelastic coefficient $\alpha_a(T)$ vs. *T* (inset). **d** Isothermal transverse strain $\varepsilon_t$ vs. **H** parallel to [111] at *T* = 2 K (*orange*) and 34 K (*green*). **e** $\varepsilon_a(H = 15\,T)$ (*blue*) and $\varepsilon_t(H = 15\,T)$ (*red*) vs. *T*. This *panel* shows a field-induced broken fcc symmetry with a different Poisson ratio and an inverted sign in the PM and AFM phases, a consequence of the strong linear term $\alpha(T)$. This is also visualized by *yellow rhombuses* in the figure. *Arrows* mark the direction of applied magnetic field. The axial magnetostriction was also measured in the paramagnetic state on a different sample with **H** parallel to [100], not shown, and found to follow an $H^2$ field dependence with a magnitude ~5× smaller than along [111]. Indications of irreversibility were found in the magnetostriction data in the AFM state, indicated with an *arrow* in **c**

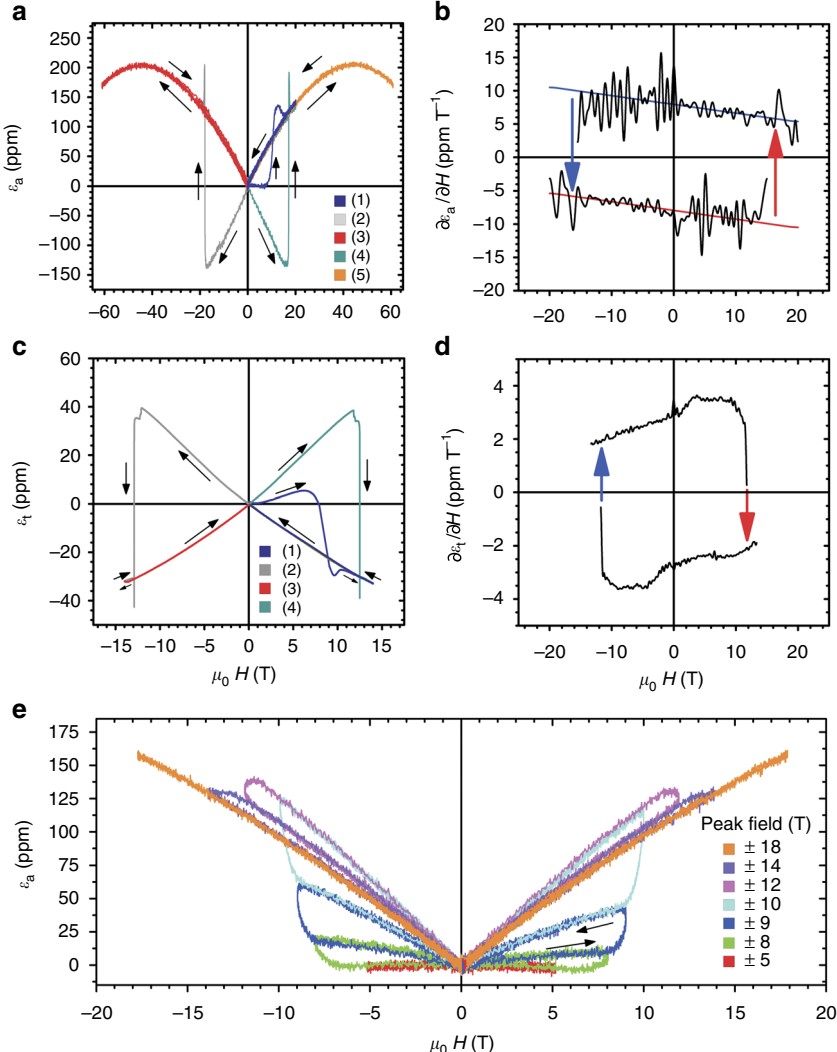

**Fig. 2** Domain dynamics and magnetoelastic memory in UO$_2$. **a** Axial magnetostriction $\varepsilon_a$ vs. **H** parallel to [111] in the AFM state of UO$_2$ in pulsed magnetic fields ($T = 2.5$ K). The measurement sequence is explained in the text. **b** Magnetostriction slope $\partial\varepsilon_a/\partial H$ vs. magnetic field parallel to [111], along a strain hysteresis loop showing remanence. *Lines* are fits that reveal $y$-axis intersections $\alpha_a = \pm 10.5$ ppm T$^{-1}$ and slope $2\beta_a = 0.17$ ppm T$^{-2}$. **c** Transverse magnetostriction $\varepsilon_t$ vs. **H** parallel to [111] measured at $T = 2.2$ K in a superconducting magnet using a sequence like the one used in **a**. Here we observe qualitatively similar, yet opposite in sign, behavior as in $\varepsilon_a(H)$. **d** $\partial\varepsilon_t/\partial H$ corresponding to the hysteresis loop in **c** showing remanence. The fitted $y$-axis intersections are $\alpha_t = \pm 2.98$ ppm T$^{-1}$, and slope $2\beta_t = 0.09$ ppm T$^{-2}$. **e** A partial domain reorientation effect, magnetoelastic butterfly, is obtained when magnetic fields are pulsed consecutively to fields between 5 T (*red*) and 14 T (*purple*). An equilibrium state is achieved with a 18 T pulse (*orange*). As seen, the domain reorientation effect can be partial, allowing for tuning of $\partial\varepsilon/\partial H$. These characteristics make the gradual reorientation of magnetic domains a peculiar memory effect in UO$_2$

known[14, 16, 18–21]. These, together with previously unseen field-induced broken cubic symmetry and memory effects in UO$_2$, are likely related to complex magnetoelastic properties important for both applied and fundamental aspects. We propose a model Hamiltonian that is capable of reproducing the main experimental features.

## Results

**Axial and transverse MS.** The MS, $\varepsilon = \Delta L/L$ (ppm), where $\Delta L$ is the sample length change with respect to the original length $L$, of UO$_2$ was measured with applied magnetic fields up to 92.5 T along the [111] crystallographic direction, at various temperatures. In the paramagnetic (PM) state ($T \geq T_N$) the axial MS $\varepsilon_a(H)$, displayed in Fig. 1c, is negative and proportional to the square of the magnetic field. Upon cooling into the AFM state ($T < T_N$) in zero field, however, an additional positive linear term abruptly appears in the MS above a critical applied field. The MS

can then be described by the expression $\varepsilon_a(H, T) = \alpha_a(T)H + \beta_a H^2$, with $\alpha_a(T) \geq 0$ and $\beta_a < 0$ for positive fields. The temperature dependence of the linear term $\alpha_a(T)$, Fig. 1c *inset*, shows characteristics of an order parameter. The quadratic term $\beta_a$ is essentially constant in the entire experimental temperature range and is likely a consequence of the Zeeman effect on the U atom's $\Gamma_5$ triplet ground state, whose degeneracy is split into three singlets by the presence of a molecular field in the 3-**k** state. Above $T_N$ the results by Caciuffo *et al.*[22] show that, even in the PM state, the triplet is split into three singlets, suggesting uncorrelated 1-$k$ dynamic Jahn-Teller distortions. A peculiar irreversible anomaly in the MS was observed when cooling the sample down in zero field (ZFC) and then sweeping the field past 18 T (*see arrow*). This behavior is described in more detail below. Additional MS data, taken in transverse geometry in a superconducting magnet to 15 T, Fig. 1d, show similar functional form $\varepsilon_t(H, T) = \alpha_t(T)H + \beta_t H^2$ with $\alpha_t(T) \leq 0$ and $\beta_t > 0$. Taken

together, $\varepsilon_a(H, T)$ and $\varepsilon_t(H, T)$ indicate a strong field-induced trigonal distortion of the zero field cubic lattice structure that changes sign upon entering the AFM state. When the axial and transverse strain at a constant field $\varepsilon_{a,t}(H = 15\,T, T)$ is considered, we notice in Fig. 1e that the onset of linear MS carries also a change in the Poisson ratio ($\nu = -\varepsilon_t/\varepsilon_a$) from 0.4 in the PM state to 0.23 below $T_N$.

The ZFC axial MS measured at T = 2.5 K is shown again in Fig. 2a, alongside additional field sweeps identified with numbers. We see in trace 1 (*dark blue*) a very small $\varepsilon_a(H)$ signal to $H \approx 10\,T$ on the field upsweep, but then increases with increasing field, going through a local maximum at $H \leq 20\,T$. During the field down-sweep the strain follows a monotonic linear decrease to zero with no visible remanence. Trace 2 (*light grey*) was measured in a subsequent negative magnetic field pulse, and $\varepsilon_a$ was observed to turn negative, displaying a minimum and a rapid switch to positive values at approximately $-18\,T$. Again, a monotonic decrease to zero strain with no remanence is observed during the field down-sweep. Trace 3 (*red*) was measured during a second negative pulsed field, and a clear $\alpha_a H + \beta_a H^2$ with $\alpha_a, \beta_a < 0$ behavior is observed. That is, the same dependence on magnetic field shown in Fig. 1c (T = 2.2 K) but with a negative linear term is observed. When the field direction is changed once again, trace 4 (*dark cyan*) is obtained, displaying a minimum at 18 T and a rapid switch to positive values, with a monotonic decrease to zero as $H$ is swept back to zero. Finally, when a second consecutive positive field is pulsed, trace 5 (*orange*), the strain is again a quadratic function of the magnetic field with $\alpha_a > 0$ and $\beta_a < 0$.

We do not see any measurable remanence in the sample length on removal of the external field. There is, however, remanence in the response rate of the lattice, i.e., $\partial\varepsilon/\partial H$. Figure 2b shows $\partial\varepsilon_a/\partial H = \alpha_a + 2\beta_a H$ vs. $H$, computed from data in Fig. 2a. The area enclosed represents work performed by the magnetic field. Similar results (save opposite signs) were found for the strain measured perpendicular to the applied magnetic field, $\partial\varepsilon_t/\partial H = \alpha_t + 2\beta_t H$, in a superconducting magnet, and displayed in Fig. 2c, d. The field-independent terms $\alpha_a$ and $\alpha_t$ are related to the components of the magnetoelastic tensor $\boldsymbol{\Lambda}$, as shown in the Supplementary Note 1. The expected theoretical ratio is $\alpha_a/\alpha_t = -2$ to be compared with the experimental ratio of $10.5/(-2.98) \approx -3.52$ (*upper branch* in Fig. 2b, *lower branch* in Fig. 2d). By symmetry considerations, the tensor $\boldsymbol{\Lambda}$ for UO$_2$ has only three identical non-zero components ($\Lambda_{14}$) whose magnitude is $10^{-9}\,Oe^{-1}$, comparable to some of the highest known[14, 18–20]. $\Lambda_{14}$, and consequently the $\alpha$'s, are odd under time reversal, strongly suggesting that the system switches between magnetic states related by time reversal. The memory effect in the switching dynamics was investigated further by applying magnetic field pulses smaller than that necessary to achieve the equilibrium state. Immediately after cooling the sample in zero field, we show in Fig. 2e how a partial jump in $\varepsilon_a(H)$ is obtained when magnetic field pulses of increasing magnitude were applied. This sequence was repeated for negative field direction. The reversible curve 1 (*red*) was obtained with a 5 T pulse. Irreversible curves, 2 (*green*) through 6 (*purple*), were obtained with peak fields between 6 and 14 T. Finally, the reversible curve 7 (*orange*) obtained with a pulse up to 18 T shows the equilibrium state. It is important to note that the envelope defined by these data sets follows very closely curve (1) in Fig. 2a, with smaller coercive field. We noticed that the fastest field sweep results in a higher coercive field (see Supplementary Note 3).

The interaction between magnetic field and AFM order in PZM systems, considered before by Scott and Anderson[23] in the context of magnetite[18], only allows linear coupling when time-reversal symmetry is non-trivially broken. Indeed, while all magnets break time reversal symmetry, AFMs fall into two categories: those where a symmetry element of the lattice can restore the original state after the $t \rightarrow -t$ transformation, and those where no such symmetry element exists in the lattice. The 3-**k** order in UO$_2$ displayed in Fig. 1b (space group $Pa\overline{3}$, point group $m\overline{3}$) belongs to the latter[3, 24–26] and hence allows a linear term in the MS. It has been postulated that the uranium 5$f$ ordered magnetic moment of 1.74 $\mu_B$[27], strongly reduced from the 3.2 $\mu_B$ value expected for the $J = 4$ multiplet, is an indicator of the importance of combined crystal electric field effects and the Jahn-Teller coupling in UO$_2$[28]. These effects substantiate the possibility of a strong coupling between external fields and the U-atom environment[12, 29], breaking the degeneracy between states connected by a time-reversal transformation.

One of our most remarkable findings is that UO$_2$ shows a non-zero $\alpha_a$, such that reversing the direction of the applied field changes the linear trigonal distortion from extension to compression until the switching field is exceeded, at which point the trigonal extension is recovered. A similar strain hysteresis or 'butterfly' memory loop, with significantly smaller switching fields, occurs in DyFeO$_3$[20], where the switching is achieved via a rotation of the AFM vector (defined as the sum of U moments on face centers minus the moment on U corner in the U unit cell) between two equilibrium states connected by time reversal. The dependence of the MS tensor on the sign of the AFM vector causes the sign change of $\alpha$. We think the mechanism at play in UO$_2$ is similar, with magnetocrystalline anisotropy creating an energy barrier between two equilibrium states with opposite AFM vector. Indeed, when UO$_2$ is ZFC-cooled below $T_N$, a static Jahn-Teller distortion of the oxygen cage takes place, corresponding to a trigonal distortion compatible with the 8c Wyckoff position of the cubic space group (205) $Pa\overline{3}$ (Fig. 1a, b). The consequent magnetic anisotropy generated by the distortion stabilizes a 3-**k** transverse order ($T_A$ in the notation described by Santini[6]). As there are four different possible Jahn-Teller distortions, each one along a different cube diagonal, there are a total of eight possible degenerate 3-**k** magnetic orders. Four of them have one of the four U magnetic moments along the [111] direction, and the other four are their time-reversed states. From a magnetic point of view, there are thus only two domains with opposite AFM vector. At zero magnetic field, the eight degenerate domains can in principle coexist. When a positive magnetic field is applied along the [111] direction, the subset of four 3-**k** domains with positive AFM vector is energetically favored. A negative [111] magnetic field favors domains with negative AFM vector. The same argument is valid for the structure labeled $T_B$ by Santini[6]. Our data in Fig. 2e suggest that the ZFC selection/ rotation of domains happens gradually as the magnetic field is increased towards ~18 T. Once this high-field state is established, the domains remain stable even after complete removal of the magnetic field as demonstrated by the history dependence in the MS. The traces obtained, when plotted together, make a magnetoelastic butterfly evident. The lack of remanence mentioned above is then not surprising, as all states share the same zero-field lattice parameter $a$. If domain boundary effects are neglected, a change in the number or distribution implies neither contraction nor expansion of the sample.

**Magnetization vs. magnetic field**. Besides MS we also measured the magnetization of a UO$_2$ crystal with **H** parallel to [111] (see Fig. 3a). The magnetic moment induced at 60 T, 0.5–0.6 $\mu_B$ per U, is still far from the estimated saturation value of 1.7 $\mu_B$ per U[27]. Figure 3b shows the magnetization measured in positive and negative magnetic fields. When the dominant linear contribution is subtracted, a more complex structure

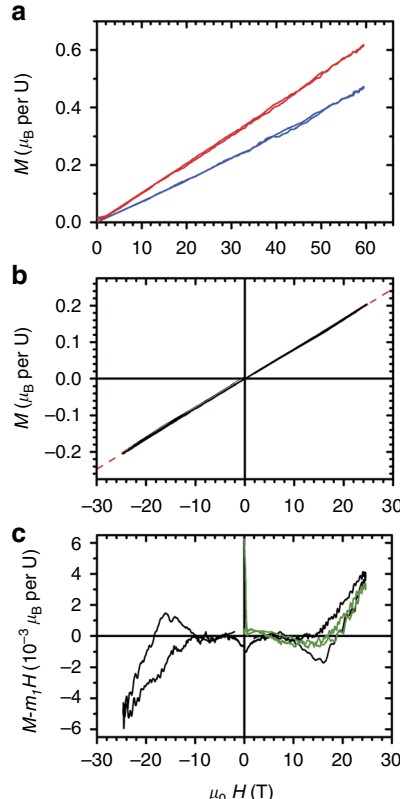

**Fig. 3** Magnetization versus magnetic field in $UO_2$. **a** Magnetization $M$ vs. magnetic field $\mu_0 H$ measured in a pulsed magnet to 60 T, with **H** parallel to [111], in the AFM (*blue curve*, $T = 4$ K) and paramagnetic (*red curve*, $T = 35$ K) states. Besides a reduction in the slope at low temperatures, very little structure is observed. The maximum induced moment is 0.5–0.6 $\mu_B$ per U. **b** Magnetization vs. field measured to 20 T ($T = 4$ K) for both positive and negative magnetic fields. No structure or irreversibility is evident to the naked eye. The *red dashed line* is a linear fit to the data, with slope $m_1 = 7.12 \times 10^{-3}\,\mu_B T^{-1}$ per U. **c** When a linear contribution is subtracted, some structure becomes apparent. *Black lines* depict the resultant magnetization on the first field pulse after cooling the sample in zero field in each direction. *Green lines* depict the data obtained on a second pulse applied in the same direction

becomes apparent. *Black lines* in Fig. 3c depict the magnetization after subtraction of a linear term on the first field pulse in each direction, right after ZFC-ing the sample. *Green lines* illustrate the data obtained on a second pulse in the same direction. A hysteretic behavior as well as a shoulder in the data at $H = 18$ T are clearly seen, which coincides with $H_{coer}$ seen in the MS. The net magnetization at the shoulder might be consequence of a broken crystal symmetry in response to external fields.

**Theoretical model.** To model and understand the observed properties in $UO_2$ we consider a minimalistic classical Hamiltonian where the degrees of freedom are the orientation of the magnetic moments at the four U atoms (4a positions) in the $Pa\bar{3}$ unit cell. They are described by their projections along the cartesian coordinates $\hat{\mathbf{S}}_i = [\sin\theta_i \cos\phi_i \hat{\mathbf{x}} + \sin\theta_i \sin\phi_i \hat{\mathbf{y}} + \cos\theta_i \hat{\mathbf{z}}]$ (see also Supplementary Note 2). The Hamiltonian includes: a Zeeman term that takes into account the interaction with the external field:

$$H_Z = -g\mu_B S_0 \overline{H} \cdot \sum_{i=1}^{4} \hat{\mathbf{S}}_i; \tag{1}$$

a magnetic anisotropy term stabilized by the static Jahn-Teller distortion of the oxygen cage:

$$H_A = -A S_0^2 \sum_{i=1}^{4} \left(\hat{\mathbf{S}}_i \cdot \hat{\mathbf{v}}_i\right)^2, \tag{2}$$

where $\hat{\mathbf{v}}_i$ are the unit vectors along the local anisotropy directions $(1,1,1)$, $(\bar{1},\bar{1},1)$, $(\bar{1},1,\bar{1})$, $(1,\bar{1},\bar{1})$ for the four U atoms, such that their vector sum is zero; a Heisenberg interaction compatible with the symmetry operations of the magnetic group:

$$H_{SS} = -4J \sum_{1 \le i < j \le 4} S_{ix}(\hat{\mathbf{v}}_i)S_{jx}(\hat{\mathbf{v}}_j) + S_{iy}(\hat{\mathbf{v}}_i)S_{jy}(\hat{\mathbf{v}}_j) + S_{iz}(\hat{\mathbf{v}}_i)S_{jz}(\hat{\mathbf{v}}_j), \tag{3}$$

where $S_{ir}(\hat{\mathbf{v}}_i)$ $(r = x, y, z)$ are the three components of the magnetic moments with the z component along $\hat{\mathbf{v}}_i$[10, 11]. Here the sum accounts for the interactions between the four magnetic moments in the simple cubic unit cell and the factor four for the interaction with their images due to the periodic boundary conditions, the elastic energy of the cubic crystal:

$$H_{el} = \frac{a^3}{2}\Big[c_{11}\left(\epsilon_{xx}^2 + \epsilon_{yy}^2 + \epsilon_{zz}^2\right) + 2c_{12}\left(\epsilon_{xx}\epsilon_{yy} + \epsilon_{xx}\epsilon_{zz} + \epsilon_{yy}\epsilon_{zz}\right)$$
$$+ c_{44}\left(\epsilon_{xy}^2 + \epsilon_{xz}^2 + \epsilon_{yz}^2\right)\Big], \tag{4}$$

the magnetoelastic energy:

$$H_{me} = -E\left[\epsilon_{yz}H_x + \epsilon_{xz}H_y + \epsilon_{xy}H_z\right]M_{st} \tag{5}$$

proportional to the staggered magnetization

$$M_{st} = \sum_{i=1}^{4} \hat{\mathbf{S}}_i \cdot \hat{\mathbf{v}}_i, \tag{6}$$

which is at the root of the PZM (see Supplementary Notes 1 and 2). At zero magnetic field, the anisotropy and Heisenberg terms are the only contributions to the Hamiltonian. They are responsible of the stabilization for the 3-**k** AFM order.

The parameters $A$, $J$, $E$, and $c_{44}$ were obtained following the criteria described in the Methods section below. The result of minimizing the total energy at $T = 0$ in this model are shown in Fig. 4, where the energetic stabilization of the magnetic domains with opposite AFM vector for positive and negative fields applied along [111] can be seen in Fig. 4a. The dependence of the deformation with the magnetic field is obtained from the energetic model by computing the derivative of the total energy with respect to the shear components of the strain tensor, with the result

$$\epsilon_{xy} = \frac{E}{c_{44}a^3} M_{st} H_z = \Lambda_{14} H_z \tag{7}$$

with similar expressions for the other strain components showing the linear dependence of the deformation with the magnetic field. For a magnetic field along the [111] direction, the linear term

$$\alpha_a = \frac{2E}{\sqrt{3}c_{44}a^3} M_{st} \tag{8}$$

changes sign with the staggered magnetization $M_{st}$ (Eq. (6)). The magnetization and deformation vs. magnetic field are shown in Fig. 4b, c, respectively. The model successfully reproduces the physics observed quantitatively, with a reasonable agreement for the MS slope and magnetization values, although the curvature in $M(H)$ is more pronounced than in the experiment. It is notable that a quadrupole–quadrupole term is not required in our

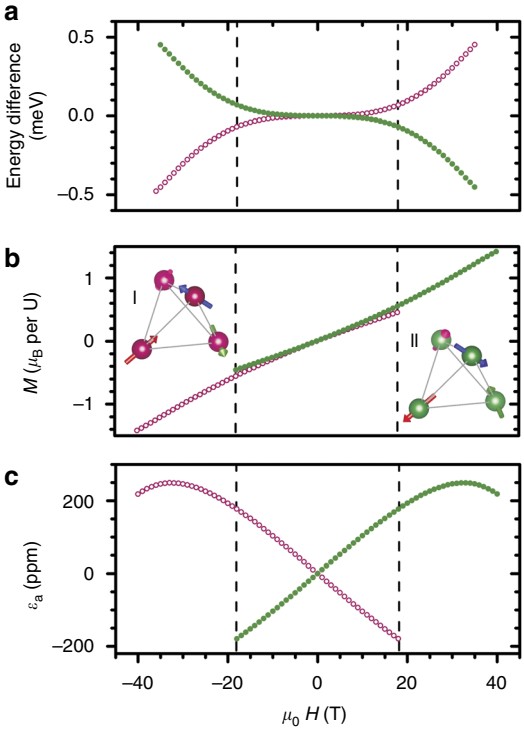

**Fig. 4** Model results. **a** Energy difference vs. magnetic field $\mu_0 H$ applied along [111] calculated with our model Hamiltonian at $T = 0$ for two magnetic states of the 3-**k** structure connected by time reversal. While the two states are degenerate in zero field, a sign-dependent difference builds up as the field is increased. **b** Magnetization $M$ vs. field, showing a relatively small yet finite difference between states. The *dashed lines*, placed at the experimental switching fields of $\pm 18$ T, are guides to the eye. **c** Magnetostriction vs. field, showing large difference between states connected by time reversal in very good agreement with experimental results. The insets **I** and **II** show the two magnetic states of the 3-**k** structure connected by time reversal. The color of the four U atoms in the unit cells (*green* or *red*) matches the corresponding color of the *curves*. The essential aspects of the experimental results are reproduced

computations. The switching fields, with only the Zeeman terms present, however, are not reproduced in the model. As is well known[6], an auxiliary quadrupolar order parameter is thought to drive the first-order nature of the AFM transition in zero field. One possibility is that piezomagnetic strain enhances the quadrupolar interaction strength, and causes the system to transition through the quadrupolar phase[9, 30] at a certain value of the trigonal compression. Since the switching field varies depending on history and field sweep rates, crystal defects and impurities could also play a role. These lines of research are, however, beyond the scope of the present discussion and will be explored separately.

## Conclusions

We have completed the first low-temperature MS study of UO$_2$ in magnetic fields up to 92.5 T, and uncovered a linear dependence on $H$ consistent with predictions[13] based on the non-collinear 3-**k** magnetic order that breaks time-reversal symmetry in a non-trivial way. This low-symmetry state is the cause of PZM in UO$_2$. The record high coercive field of 18 T makes it a piezomagnet of formidable hardness[14, 16, 18–21]. We propose a model Hamiltonian that is capable of reproducing the main experimental features, and points to the importance of a competition between exchange interactions and magnetic

anisotropy. The unusually low thermal conductivity of UO$_2$ cripples its performance as a fuel in nuclear reactors[31]. Here we uncover first-order coupling between the magnetism in U-atoms and lattice degrees of freedom that could be the origin of scattering of phonons against spin fluctuations dressed with dynamic Jahn-Teller oxygen modes[22] well above $T_N$. These effects should be explored further. PZM, the magnetic counterpart to piezoelectricity[32], is also a property currently being discussed as a strategy to control magnetism by electricity[16, 21] at the nanoscale. Our results and modeling on UO$_2$, applicable to other piezomagnets, could have an impact on current efforts in this direction. Inelastic neutron scattering as well as X-ray scattering experiments in high magnetic field are planned to further test the details of the field-induced broken symmetries revealed by the results presented here.

## Methods

**Experimental**. Several single-crystal samples of UO$_2$ were X-ray-oriented, and cut in the shape of mm-long bars, each along a different principal crystallographic axis ([100], [110], and [111]). Variations in the sample length $L$ as a function of the temperature and/or magnetic field $\Delta L/L = [L(H, T) - L(H_0, T_0)]/L(H_0, T_0)$ were measured using a fiber Bragg grating (FBG) technique[33–35] consisting of recording spectral information of the light reflected by a 0.5 mm long Bragg grating inscribed in the core of a 125 µm telecom-type optical fiber. The FBG section of the fiber is glued to the sample to be studied, and changes in the grating spacing are driven by changes in the sample dimension $L$ along the fiber when temperature or magnetic field is changed. We use here the definition $\varepsilon = \Delta L/L$, in units of parts per million (ppm). Capacitor bank-driven pulsed magnets were used to produce magnetic field pulses to 60 T, and a 100 T repetitive pulse magnet energized by a motor generator and a capacitor bank was used up to 92.5 T[34, 35]. Owing to sample space limitations in pulsed magnets, the transverse MS was measured in a superconducting magnet furnished with a $^4$He flow cryostat.

**Computational**. We used the Broyden–Fletcher–Goldfarb–Shanno algorithm as implemented in the open source SciPy package (http://www.scipy.org) for the energy minimization at $T = 0$. The anisotropy $A = 2$ meV and exchange $J = 0.8$ meV values are found necessary to obtain the experimental critical temperature $T_N = 30.8$ K using a standard Metropolis Monte Carlo algorithm at $H = 0$. A magnetoelastic interaction $E = 0.280$ meV T$^{-1}$ is used to match the experimental value $\varepsilon_a(H = 20\,\text{T}) = 210$ ppm. Other parameters used are the experimental shear elastic constant $c_{44} = 60$ GPa, the experimental lattice parameter $a = 5.47$ Å, $g = 2$, and $S_0 = 1$.

**Data availability**. All data generated in this study are available from the authors upon request.

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

## Acknowledgements
Work by K.G. was supported by the U.S. Department of Energy, Office of Basic Energy Sciences, Materials Sciences, and Engineering Division project "Actinide materials under extreme conditions". The NHMFL Pulsed Field Facility is supported by the NSF, the U.S. D.O.E., and the State of Florida through NSF cooperative grant DMR-212 1157490. Work by N.H. and M.J. was supported by the U.S. D.O.E. BES project "Science at 100 Tesla". We deeply thank the NHMFL 100 T magnet operation team: Y. Coulter, D. Roybal, M. Gordon, J. Martin, and H. Teshima. We thank A. Shethker, C.D. Batista, and J.F. Scott for their interest in our results and insightful discussions. A.S. thanks fruitful discussions with Céline Varvene and Jaejun Yu. M.B.S. gratefully acknowledges his visiting scientist status at NHMFL.

## Author contributions
K.G. proposed the experimental studies. M.J. and K.G. designed the research. M.J, V.S.Z., N.H., and K.G. performed the experiments. T.D., J.C.L., C.S., D.A.A., and J.L.S. provided the UO₂ single crystal samples. A.S., M.J., and M.S. discussed and developed the theoretical model. A.S. performed the calculations. All authors discussed and interpreted the results, and contributed to the preparation and writing of the manuscript.

## Additional information

**Competing interests:** The authors declare no competing financial interests.

