## [Peer Review File · Nature Communications]

REVIEWERS' COMMENTS:

Reviewer #1 (Remarks to the Author):

The revised version of the article by Jaime et al. is considerably improved with respect to the original version submitted to [Redacted]. The authors have addressed in a satisfactory way all the points I raised in my previous report. The addition of a theoretical analysis, although based on a simple model that takes into account only the dominant interactions, supports the interpretation of the experimental results and its validity. Nature Communications, moreover, seems to me a more suitable editorial collocation for this work than [Redacted]. I recommend publishing this article after polishing the text to get rid of some typos: spelling of some words at line 107 and in the last sentence of the caption of Fig. 1 of the supplementary information; Figure 1c in the main text shows ϵ_a , not ϵ_t as reported in the label of the y axis; in several places, sec is used as abbreviation of second. It should be s not sec. The author use "differential magnetization" to indicate the nonlinear magnetization. This should be corrected.

There are errors in some of the references: ref 5 (B 72), Ref. 8 (spelling of Erdős and journal abbreviation); Ref 9 (journal abbreviation); Ref 12 (journal name...niya not nuya); ref 18 (spelling of author name and Journal name: PRL not PRB; ref 31 is repeated (same as ref 8), Supplementary information ref 12 (article number).

Reviewer #2 (Remarks to the Author):

This manuscript was submitted previously to [Redacted] and I reviewed it. The experimental data are very elegant, but they previously had some problems with both the references concerning UO₂ as well as the model they proposed, which was far from rigorous.

I am pleased to say that the authors have made major revisions, especially in their model, which is now completely new, and I recommend this paper for publication without further changes.

Reviewer #3 (Remarks to the Author):

The manuscript describes investigation of piezomagnetism in uranium dioxide. It reports measurements on single crystals subjected to magnetic fields which are increased in magnitude up to 90T along [111] directions both above and below the Neel temperature, T_N , where UO₂ undergoes a first order transition from a paramagnetic (PM) to a 3k non-collinear anti-ferromagnetic (AF) state. Below T_N a positive linear magnetostriction is observed which leads to a trigonal distortion of the cubic lattice. Reversing the direction of the applied field causes a sharp transition of the strain at -18T and this switching behaviour is repeated on further cycling of the magnetic field. Axial and transverse magnetostriction are measured as functions of temperature and field along with the induced net magnetisation along the field direction. The axial strain, for example, has a simple quadratic field dependence above T_N but has an additional large linear term below.

The results are accompanied and interpreted by a theoretical model which correctly captures the principal features of the measurements. Drawing on insights from the literature the authors have constructed a model for the U spins which comprises a Zeeman term, a local magnetic anisotropy term deduced from Jahn-Teller distortion effect, a nearest neighbour pairwise spin interaction term which together with the anisotropy accounts for the 3k AF structure in zero field, an elastic energy term for a cubic material and a field-strain coupling term with coefficient proportional to the net (staggered) magnetisation. The model is elegant and contains what appears to be the smallest number of parameters necessary to describe much of the observed physics. It enables the magnetoelastic coupling to be identified neatly. The aspects which the model fails to address are also described clearly and help to highlight where further work is needed. These include the mechanisms for the first order rather than second order paramagnetic-

AF transition magnetostriction and the strain switching with field.

Piezomagnetism is attracting considerable interest currently and this paper will be very useful in this context. I recommend it be accepted in Nature Communications.

I have a couple of suggestions of queries/modifications which the authors could consider however.

1. For the magnetoelastic energy term, eq.5 on page 7, please use another label other than "M" to describe the coupling coefficient. On first glance through the paper this looked like M for magnetisation.
2. On page 7, beginning of the second paragraph "The unknown parameters..." I think it would be helpful to list them, i.e., J,A,M, elastic constants.
3. Typo "magnetoelstic" in caption to Fig.2.
4. On the ordinate axis to fig.1(c) should it be the axial strain rather than transverse strain that is marked?